# GMLM-CNN: A Hybrid Solution to SWIR-VIS Face Verification with Limited Imagery

**DOI:** 10.3390/s22239500

**Published:** 2022-12-05

**Authors:** Zhicheng Cao, Natalia A. Schmid, Shufen Cao, Liaojun Pang

**Affiliations:** 1Molecular and Neuroimaging Engineering Research Center of Ministry of Education, School of Life Science and Technology, Xidian University, Xi’an 710071, China; 2Lane Department of Computer Science and Electrical Engineering, West Virginia University, Morgantown, WV 26505, USA; 3Department of Physiology and Biophysics, Case Western Reserve University, Cleveland, OH 44106, USA

**Keywords:** cross-spectral face recognition, SWIR, measurement level, hybrid, feature fusion, limited imagery

## Abstract

Cross-spectral face verification between short-wave infrared (SWIR) and visible light (VIS) face images poses a challenge, which is motivated by various real-world applications such as surveillance at night time or in harsh environments. This paper proposes a hybrid solution that takes advantage of both traditional feature engineering and modern deep learning techniques to overcome the issue of *limited imagery* as encountered in the SWIR band. Firstly, the paper revisits the theory of measurement levels. Then, two new operators are introduced which act at the nominal and interval levels of measurement and are named the Nominal Measurement Descriptor (NMD) and the Interval Measurement Descriptor (IMD), respectively. A composite operator Gabor Multiple-Level Measurement (GMLM) is further proposed which fuses multiple levels of measurement. Finally, the fused features of GMLM are passed through a succinct and efficient neural network based on PCA. The network selects informative features and also performs the recognition task. The overall framework is named GMLM-CNN. It is compared to both traditional hand-crafted operators as well as recent deep learning-based models that are state-of-the-art, in terms of cross-spectral verification performance. Experiments are conducted on a dataset which comprises frontal VIS and SWIR faces acquired at varying standoffs. Experimental results demonstrate that, in the presence of limited data, the proposed hybrid method GMLM-CNN outperforms all the other methods.

## 1. Introduction

Recognition of individuals based on their facial appearance has been a subject of interest for many decades. Application of new imaging modalities such as near-infrared (NIR), short-wave infrared (SWIR), mid-wave infrared (MWIR) and long-wave infrared (LWIR) has recently opened up new opportunities for research in the area of IR face recognition [1], which is usually encountered in monitoring human activities at night or in harsh environments. For example, face recognition in the infrared is necessary at nighttime when visible light is infeasible to acquire a face image. In addition, IR-involved face recognition technology is preferred under bad weather conditions such as raining, fog, etc., due to the higher imaging ability of IR imaging than visible light.

IR imagery demonstrates characteristics superior to imagery acquired in the visible light band [2]. For example, thermal IR captures the patterns associated with heat emission of a subject. In addition, unlike visible light, IR is usually less susceptible to the environmental illumination or even needs no external source of illumination at all. Another advantage of IR (especially thermal IR) is its inherent ability to emphasize not only the face geometry and texture of the face as in the visible range, but also the anatomical structures beneath the skin.

IR face recognition has already attracted a lot of attention [3,4,5,6,7,8,9,10,11,12,13,14,15,16,17,18,19,20,21,22,23,24,25]. The research works can be categorized as either intra-spectral face recognition or cross-spectral face recognition. *Intra-spectral* face recognition refers to matching images within the IR band. Multi- and hyper-spectral face recognition [26] are special cases of intra-spectral face recognition where several subbands of the IR are fused to generate a multi- or hyper-spectral imagery, and then matching is performed between the fused images. *Cross-spectral* face recognition [8], on the other hand, matches IR images against visible light images. Since the imaging mechanism as well as the facial tissue reflective properties under visible light and IR are quite different, face images in the two cases demonstrate quite distinct characteristics (see the sample face images in Section 5.1). Therefore, cross-spectral face verification presents a much more challenging problem.

Addressing this problem of cross-spectral face verification would lead to many potential real-world applications, such as all-time and all-weather secure surveillance will be possible at public venues such as airports, office buildings, parking lots, etc. Solders in battlefields can wear night vision devices that will be able to automatically identify terrorists hidden in the darkness. Border patrolling officers can deploy cameras at the boarder to automatically track and alarm for suspicious trespassers during off-hours.

This work focuses on the problem of cross-spectral face verification, and more specifically, of matching face images between visible light and an under-explored IR subband—*SWIR*. The very subband of SWIR is chosen for study due to its advantages over other IR subbands such as NIR and LWIR [27]:SWIR cameras generate cleaner imagery in harsh atmospheric conditions such as rain, snow, and fog [28,29];SWIR cameras produce higher SNR images at night and are therefore more suitable for long range imaging at night [30];SWIR is invisible to the human eye and is undetectable by silicon-based cameras [30].

As a special case of cross-spectral face recognition, verification between SWIR and visible light is also confronted with the issue of feature extraction in a common space. However, this issue in the case of the SWIR band is more pronounced than other IR subbands such as NIR because SWIR images are even more distinct from visible light faces than NIR images. More importantly, current SWIR imageries are very difficult to obtain due to the cost of collection and a low popularity of usage. As a result, the sizes of publicly available SWIR face datasets are much smaller compared to dataset sizes of visible light and other IR subbands. Traditional non-deep learning methods need no large amount of training data but are lower in robustness and performance. As deep learning techniques emerge, how to address the problem of SWIR-Vis verification in the presence of *limited imagery* poses a severe challenge.

In view of the issue of limited imagery as aforementioned, this research work provides an alternative solution to traditional hand-crafted operator based methods and current deep learning methods. This paper proposes to combine a composite operator with a succinct PCA-based neural network. Inspired by the theory of measurement levels, the composite operator fuses multiple levels of measurement information to extract distinguishing features. These features are input into a following neural network that is constructed using the matrix decomposition tool of PCA, where a high-level feature learning and classification is conducted. Such a combination of composite operators and neural networks has advantages of automatic global optimization and feature robustness as deep learning methods possess, and advantages of very fewer parameters and less training requirements.

In addition to designing a hybrid method, there remains another interesting question for SWIR-Vis verification. How each measurement level contributes to the final recognition performance should be investigated. It is also of interest to find out whether recognition performance increases by simply adding more measurement levels. This paper is therefore driven to design more cross-spectral experiments to quantitatively study these questions.

The remainder of the paper is organized as follows: Section 2 provides a thorough literature review on the topic of cross-spectral face recognition. Section 3 summarizes the motivations and contributions of our work. Following introduction of the theory of measurement levels, Section 4 presents two new operators NMD and IMD, proposes GMLM as a fused operator, and finally introduces the hybrid solution GMLM-CNN. Section 5 describes the experimental setup and analyzes and compares the performance of GMLM-CNN and four other methods, as well as individual levels of measurement and their combinations. Section 6 concludes the paper.

## 2. Review of Relevant Research

Most of the face recognition algorithms in the literature were developed for the purpose of intra-spectral matching, and more specifically, for matching visible light probes to a gallery of visible light images. As the IR band attracted more attention, some operators were later on tuned to work with face recognition in application to IR face images. Matching IR face probes against an IR face gallery falls in the category of intra-spectral face recognition.

As an example of earlier related works, Pan et al. [26] collected a face dataset acquired in 31 narrow IR sub-bands ranging from 0.7 μm to 1.0 μm. They demonstrated effectiveness of a hyperspectral approach where a spectral reflectance vector evaluated in face regions at different wavelengths is employed as the feature vector. Chen et al. [31] applied PCA to study face recognition in the thermal IR and visible light bands, separately, and compared their performance with Faceit G5. Lin et al. [32] introduced a Common Discriminant Feature Extraction method that brings images from different modalities (visible light, NIR and sketches) into a common feature space. It was shown that the proposed algorithm outperforms PCA, LDA, kernel PCA, and kernel LDA in the visible versus NIR comparison and also when optical photos are matched against sketches. Li et al. [33] proposed a method to compare face images within the NIR spectral band under different illumination scenarios. Their face encoder involved the LBP operator to achieve illumination invariance and was applied to NIR images acquired at a short distance.

Another special scenario of face recognition involving IR is cross-spectral face recognition between visible light and IR face images. For example, the work of Klare and Jain [5] employed a method based on LBP and HOG features, followed by the LDA algorithm to reduce dimensionality. The method was applied to NIR and color images for cross-spectral matching. The results were shown to outperform Cognitec’s FaceVACS [34]. In their work, Kong et al. [35] performed fusion of NIR and thermal IR face images in the Discrete Wavelet Transform domain on the NIST/Equinox and UTKIRIS [36] datasets. They showed that, when the fused images are fed to the Faceit recognition software, the resulting matching performance improves with respect to the case when the same face classes are compared within the same spectral band, NIR or thermal IR in this case. Yi et al. [37] encoded images captured in NIR and visible bands by adopting a Laplacian of Gaussian (LoG) filter. The method compared common patches (partial faces) in visible and NIR images. The experiments were performed on MBGC data [38,39].

Liao et al. [40] applied a Multiscale Block Local Binary Patterns (MB-LBP) descriptor to NIR and visible face images. Both types of images were preprocessed with Difference of Gaussian (DOG) filters and then encoded with the MB-LBP operator. AdaBoost was applied to select features, and a regularized LDA method was used to match processed data. The method was tested on a multispectral dataset of 52 face classes. The implemented approach was shown to outperform CDFE, PCA-CCA, and LDA-CCA [41] methods when visible images are matched against NIR images. Akhloufi and Bendada [42] used the classical Local Ternary Pattern (LTP) and a new Local Adaptive Ternary Pattern (LATP) operator. They experimented with both Equinox and the Laval University Multispectral Database [43], involving visible, SWIR, MWIR, and LWIR. The authors conducted a multi-resolution analysis in the “texture space" to fuse images from different spectral bands and reported that the fusion of different bands improves recognition rates when images are matched within the same band.

Maeng et al. [44] reported the results of long range cross-spectral face matching, where long range NIR images are matched against visible face images. The paper introduced a new long range NIR database called Near-Infrared Face Recognition at a Distance Database (NFRAD-DB). Face recognition performance was evaluated using FaceVACS, DoG-SIFT, and DoG-MLBP methods. The experiments involved 50 long range NIR classes and more than 10,000 short range visible face images. The achieved rank-1 recognition performance was 28%. Nicolo and Schmid [6,45] explored the case of matching SWIR face images against visible light images at long standoff distances. They developed a new compound operator for this special case by utilizing both the magnitude and phase response of a Gabor filter bank combined with LBP, GLBP, and WLD. The operator outperformed Faceit G8. The results were demonstrated on two datasets consisting of visible and SWIR images at short and long standoff distances.

Bourlai et al. [7] collected a SWIR face dataset and considered the case of face verification between visible light face images and SWIR images. Multispectral fusion of different SWIR wavelengths (a collection of eight different wavelengths) was also investigated. The work of Savvides et al. [10] proposed a dictionary learning-based approach to deal with the problem of NIR-to-visible light matching. Their approach used a joint minimization of the $L_{0}$ norm to learn a mapping function between the heterogeneous images of NIR and visible light, followed by reconstruction of VIS images hallucinated from the NIR light band and vice versa.

Cao and Schmid [46] studied cross-spectral face recognition with the involvement of NIR, SWIR, MWIR, and LWIR imagery. They modified the discrete representation of LBP and WLD into a continuous representation and proposed a new composite operator. Their experiments demonstrated that the proposed composite operator outperforms individual and other composite operators available at that time. The work of Tao et al. [47] introduced a common discriminant feature approach inspired by the probabilistic LDA, where heterogeneous face images were first encoded in a way similar to LBP and then NIR-VIS data were used to train a common discriminant model which minimized the difference between heterogeneous faces. A Gaussian kernel was added to boost the performance.

To summarize, early research works until the first decade of this century have raised and studied many issues in the area of multispectral face recognition. However, all of them deal with multispectral face recognition in a case-by-case manner, and use traditional non-deep learning methods that rely on hand-crafted operators and feature extractors. These traditional methods usually demonstrate low robustness and unsatisfactory performance, despite the devoted intensive endeavors. This situation motivates and drives upcoming researchers to keep on inputting research efforts into the multispectral problem.

Most recently, due to the advances in computing technologies, the machine learning research community turned towards deep learning approaches and deep convolutional neural networks (CNN). In application to cross-spectral face recognition, several promising results have been demonstrated. The following publications have the highest relevance to the cross-spectral face recognition.

Hu et al. [14] provides a thorough literature review on the topic of heterogeneous face recognition. Sarfraz and Stiefelhagen [13] proposed to bridge the gap between thermal and visible facial modalities by means of a deep neural network. Their model learned a nonlinear relationship between visible and thermal imagery while preserving the face identity. They claimed that the proposed approach improves Rank-1 recognition results by 10% to 30%, depending on the involved database.

Oh et al. [15] developed a single hidden-layer Gabor-based network for heterogeneous face recognition. Their experiments involved the BERC visual-thermal infrared database and CASIA visual-near infrared database.

Nasrabadi et al. [16] developed a coupled deep neural network architecture that addresses the problem of matching visible face images to polarimetric images. The network architecture was developed to make a full use of polarization state information in polarimetric imagery, which is equivalent to textural and geometric information in visible imagery. Performance analysis showed promising results.

In [48], Riggan et al. synthesized visible facial images from thermal facial images, claiming that the existing face recognition software developed for visible imagery can be applied in the situation of a cross-spectral face recognition. Zhang et al. [18,49] proposed to use a generative adversarial network to synthesize a high quality visible facial image from polarimetric thermal images. They also introduced an extended polarimetric dataset of 111 subjects.

Di et al. applied a self-attention generative adversarial network [50] and Attribute Guided Synthesis [20] to synthesize thermal facial images from visible facial images. Then, deep features were extracted both from the original and synthesized data to form a compound template. Performance analysis is performed on the ARL polarimetric thermal face dataset.

Prabhu et al. [51] introduced two face recognition techniques using face image at nine distinct spectral ranging from 530 nm to 1000 nm. They involved a CNN as a feature extractor together with SVM and k-NN as a classifier.

Tan et al. [52] conducted high-resolution synthesis of NIR faces from visible light faces by a complementary combination of a texture inpainting component and a pose correction component. They demonstrate that such a combination improved heterogeneous face recognition accuracy.

In the work of [53], Gao et al. proposed a coupled attribute learning method where the relationship between face attributes and identities were considered.

In summary, deep learning-based methods have achieved great success and have replaced traditional hand-crafted methods in the area of IR and cross-spectral face recognition, due to their advantages such as automatic feature extraction, greater robustness, and higher performance. However, deep learning approaches typically demand an enormous amount of real or simulated training data, which limits their real world application in certain special situations. For instance, with several small datasets of SWIR face images available in practice, the amount of data to train a deep CNN for SWIR-Vis cross-spectral face recognition is insufficient. SWIR image datasets are usually small due to the high cost of cameras. In addition, most SWIR datasets are collected by the military, hence they are not available for public use. In the absence of a large SWIR training database, the attempt to synthesize SWIR faces from visible images is not practical or feasible either. Furthermore, the performance and scalability of these deep learning algorithms could be truly evaluated only if a large dataset of SWIR facial images were made available to the public. Thus, it is quite critical to design deep neural networks which can overcome the issue of limited training data for the case of SWIR-Vis face recognition.

## 3. Motivation and Contribution

As pointed out in the previous section, the issue of limited training data restrains typical deep learning methods from being deployed in the case of SWIR-Vis face recognition, in spite of remarkable advances in the deep learning field. In view of this, rather than designing a pure CNN, this work addresses the problem with a hybrid concept which takes advantage of both modern deep learning techniques and traditional feature engineering (i.e., hand-crafted features). The rationale behind such hybrid concept is that deep learning is advantageous in automatic global optimization and robustness against input variety, while traditional feature engineering has significantly fewer parameters and needs much lighter training workloads. Combining the two as a hybrid solution could maintain the advantages of both, and thus would lead to an improved performance than either of them.

The proposed hybrid approach, named GMLM-CNN, is built with two modules: A composite feature extractor placed at the front and a following convolutional neural networks. The hybrid method requires minimal training, has low computational cost, and yields high matching performance. Therefore, it is especially suitable for cross-spectral face verification with limited imagery (i.e., small training datasets) as well as for mobile device-based face authentication.

We will first introduce the concept of levels of measurement and then apply it as a criterion to classify popular operators used in cross-spectral face recognition into four groups. We will then propose two new operators based on the theory of levels of measurement, the Interval Measure Descriptor (IMD) and the Nominal Measurement Descriptor (NMD), operating at the interval level and the nominal level, respectively. To the best of our knowledge, there are currently no operators designed to encode face images at these two levels. It is expected that different operators acting at different levels of measurement extract complimentary information, leading to improved cross-spectral recognition performance when the information is fused.

To showcase the complimentary nature of the information extracted at different levels of measurement, this paper further proposes to fuse the features extracted by the two new operators, NMD and IMD, with features acquired by two other operators at the remaining two levels of measurement (the ordinal and ratio levels). The composite operator that extracts all these features and fuses them is named the Gabor Multiple Level Measurement (GMLM). To illustrate the performance of GMLM, it is compared with other well-performing operators introduced in the literature. Both individual and composite operators are involved and analyzed for the cross-spectral verification performance. Experiments are conducted on two heterogeneous datasets, each composed of visible light and SWIR faces at both short and long standoffs. To acknowledge the recent advances in deep learning-based techniques, the proposed GMLM-CNN is further compared with two typical CNN models of cross-spectral face recognition, Deep Coupled Learning [16] and Attribute Guided Synthesis [20], which represent the state of the art. Additional experiments are designed to investigate the contribution of each measurement level and effects of different fusing schemes.

Our research and development led to the following main contributions:The problem of cross-spectral face verification is studied involving a relatively new IR subband—SWIR;Two new operators, IMD and NMD, are proposed which extract face image features at two different levels of measurement;A composite operator, GMLM, is introduced by fusing individual local operators acting at four distinct levels of measurement;To alleviate the issue of limited imagery, in addition to data augmentation and transfer learning, a hybrid framework of combining hand-crafted features and deep learning is introduced, where GMLM features are input into a subsequent network, which is succinct and efficient. The hybrid method is proved to achieve the state of the art;The contribution of each of the four levels of measurement is investigated. The effect of different fusion schemes is also studied.

## 4. Proposed Methodology

Due to limited imagery and training data, existing deep learning models can not be directly deployed to solve the SWIR-Vis face verification problem. One potential approach to this problem is to combine and take advantage of both traditional methods and modern deep learning techniques, that is, a hybrid solution. In this work, rather than designing pure neural networks, we propose to use hand-crafted features as the input to the following simple-structured CNN together to accomplish SWIR-Vis face verification.

In order to extract informative and distinguishing features as effectively as possible, we first revisit the *theory of measurement levels* and introduce two new feature extraction operators (NMD and IMD). We further extract face features at four different measurement levels and fuse them into a single composite feature vector (GMLM). Following this, the GMLM vector is formed as an input to a PCA-based succinct network, which finally performs the verification task. The overall hybrid method is termed GMLM-CNN. Our choice of the PCA structure is to maximally simplify the CNN. As a result, the proposed approach has dramatically fewer parameters that need to be estimated by training, distinguishing it from other deep learning methods that require a substantial amount of data for training.

### 4.1. Theory of Levels of Measurement

Level of measurement, also known as scale of measure, refers to the nature of information within the values assigned to a variable to be measured (certain attributes) and the relationship among the values. Psychologist Stanley S. Stevens proposed a typology—the best known one—with four levels for different types of measurement: the nominal, the ordinal, the interval, and the ratio levels [54].

The nominal level of measurement is often referred to as the qualitative level, while measurements made at the other three levels are called quantitative data. The concept of levels of measurement has been extensively used in various disciplines such as natural sciences, linguistics, and political sciences: taxonomic ranks in biology, parts of speech in grammar, and political affiliation in politics. As an example, Figure 1 defines the nominal level of measurement in gender classification of individuals.

The nominal level, which is also the lowest level, simply “names" the attributes of a variable to be measured uniquely by assigning certain numerical values. Ordering of the attributes is neither implied nor are arithmetic or logical operations on the assigned values meaningful. In the example shown in Figure 1, the measured variable is the gender of human, which can be female or male. The numerical value assigned is either 0 or 1. When comparing the gender of two individuals (humans), the only meaningful relationship between their nominal levels is either being the same or not.

In the case of the ordinal measurement, however, the attributes can be ranked. Larger values suggest a greater amount of a certain quality. Nonetheless, the distance (the difference) between values can not be defined. Therefore, the interval between any two values is not interpretable in such an ordinal measure. Take as an example the ranking of athletes at competitions. A place/position is assigned based on the performance of an athlete. In this example, only the ordering information is meaningful. By looking at the award, one can determine the placement of the athlete, but it is impossible to say how much better this athlete performed compared to his/her peers.

As for the interval measurement, not only the order but also the distance between attributes is meaningful. Temperature reading is a good example for interval measurement, where a higher temperature suggests a warmer weather than that of a lower temperature. The larger the temperature difference, the degree of warmness increases. However, calculating the ratio of temperatures would not make sense.

Finally, in the ratio measurement, ordering, distance, and ratio are all meaningful, and there is always an absolute zero defined. This implies that one can construct a meaningful fraction (or ratio) with a ratio variable. This level of measurement is frequently encountered in practice. This level of measurement is also commonly used in science. A few most common examples are measuring length, mass, force, etc., where the scientists will find or derive equations involving complex calculations using these variables.

It can be easily observed that the four levels of measurement have increasing complexity from the nominal to the ratio, and that they extract different types of information. Each level can be described by different mathematical or logical operations. As a result, operators and descriptors developed for face recognition applications can be partitioned into four groups (corresponding to the four levels of measurement) by inspecting the essential mathematical or logical operations involved. For instance, the LBP operator can be placed in the category of the ordinal level of measurement, since the core operation involves the determination of the order between a central pixel *x* and its neighboring pixels $x_{i}$. For HOG, the essence is to find the magnitude and orientation of the gradients, which involves multiplication and division. Therefore, it can be viewed as an operator acting at the ratio level of measurement. Another example of operators acting at the ratio level is WLD (refer to Equation (Equation 11) for details), since not only operations of difference but also operations of division are involved.

Table 1 summarizes properties of the four levels of measurement and presents examples of corresponding operators acting at different levels of measurement that are used in face recognition.

### 4.2. Nominal Measurement Descriptor

This section introduces a new operator acting at the nominal level of measurement and is thus named *the Nominal Measurement Descriptor* (NMD). To the best of our knowledge, this is the first operator reported at this level for face recognition applications.

Given an input image and a set of intensity classes that each pixel can belong to, NMD compares the class of a pixel with the class of its neighbors and encodes this information. For example, assume two intensity classes $c_{0}$ and $c_{1}$. Each pixel belongs to either of the classes, defined as
(1)$$C\left(x\right)=\left\{\begin{array}{cc} c_{0}, & x\in [0,127],\\ c_{1}, & x\in [128,255]. \end{array}\right.$$

Then, the nominal relationship between each pixel and its 8-pixel neighborhood is considered. A binary value of 0 is recorded if the classes of the central pixel and a neighbor are identical, whilst a value of 1 is recorded if there is a difference. The comparison is performed pixel-by-pixel within the 8-neighborhood. The final step of the encoding concatenates all binary values into a single binary string. Mathematically, NMD is described as
(2)$$NMD\left(x\right)=\sum_{i=1}^{8}nom\left(C\left(x\right),C\left(x_{i}\right)\right)\cdot 2^{i},$$
where $x_{i}$ is a neighbor of the central pixel *x* in the input image, and C(·) is a class assigning function. In a two-class case, it is given by (Equation 1). In general, the range of the mapping function C(·) can include up to 256 discrete classes. The operator nom(·) generates the final numerical value of nominal measurement between two pixels with assigned classes $c_{i}$ and $c_{j},$
(3)$$nom\left(c_{i},c_{j}\right)=\left\{\begin{array}{cc} 0, & c_{i}=c_{j},\\ 1, & c_{i}\neq c_{j}. \end{array}\right.$$

An illustration of the encoding with the proposed NMD descriptor is shown in Figure 2a. Examples of encoding with application to an actual face image are shown in Figure 2b–e, in which an input face image acquired at SWIR 1.5 m and the same image after filtering with a Gabor operator are encoded using the newly introduced NMD operator.

### 4.3. Interval Measurement Descriptor

The literature does not appear to employ an operator acting at the interval level of measurement. Therefore, we propose a new descriptor operating at this level. It is named *the Interval Measurement Descriptor* (IMD).

Given an input image and a neighborhood of a pixel within the image, IMD encodes the difference between the intensity of each pixel and its neighbors. Once again, a neighborhood of eight pixels is considered. The range of the difference in intensity is divided into *K* intervals. The partition can be uniform or non-uniform. An illustration of a *K*-interval partition scheme is provided in Figure 3. Each interval is further assigned an integer code. The final code is generated by concatenating the codes of individual pairs of pixels within their neighborhood (i.e., the 8 adjacent pixels). Mathematically, IMD is described as
(4)$$IMD\left(x\right)=\sum_{i=1}^{8}S\left(x_{i}-x\right)K^{i},$$
where $x_{i}$ is the *i*-th neighbor of the central pixel *x* in the input image, and *K* is the total number of intervals involved.

Denote the difference between two pixels as Δz, and let S(Δz) be the mapping between the interval that Δz falls into and the corresponding code (an integer value). The expression of S(Δz) is dependent on the partition scheme of the intervals. As an example, uniform partition is an easy and straightforward one. However, such a uniform partition is relatively naive and fails to consider the actual intensity distribution of SWIR and Vis face images. Thus, this paper proposes a *non-uniform* partition scheme based on the Shannon encoding rule. Given *K* intervals, $I_{k}$, k=1,2,…,K, let the probability of intensity that falls in each interval be $P(I_{k})$. Arrange the sequence of $P(I_{k})$ in decreasing order, i.e., $P\left(I_{1}^{{}^{\prime }}\right)\geq P\left(I_{2}^{{}^{\prime }}\right)\ldots \geq P\left(I_{K}^{{}^{\prime }}\right).$ By the Shannon’s encoding rule, the partition $\Delta \xi_{i}$ is calculated as:(5)$$\Delta \xi_{i}=-\log P\left(I_{i}^{{}^{\prime }}\right)\;,i=1,2,\ldots ,K,$$
where $\Delta \xi_{i}=\xi_{i}-\xi_{i-1}$ and $\xi_{i}=0.$ Given such a relationship, the right end of the *i*th interval $\xi_{i}$ can be in turn calculated as:(6)$$\xi_{i}=\sum_{j=1}^{i}\Delta \xi_{j}+\xi_{0}.$$

To normalize the partition to be between −255 and 255 along the horizontal axis and keep it as an integer, the new partition $\zeta_{i}$ is further calculated as:(7)$$\zeta_{i}=\left\lfloor 510\cdot \frac{\xi_{i}}{\sum_{j=1}^{K}\Delta \xi_{j}}-255\right\rfloor ,$$
where ⌊·⌋ denotes the round down operation. The new mapping function of S(Δz) using the non-uniform partition scheme is thus given by:(8)$$S\left(\Delta z\right)=k,\mathrm{if}\;\zeta_{k-1}\leq \Delta z<\zeta_{k}.$$

The non-uniform partition scheme is illustrated in Figure 3.

It is worth noting that the IMD operator is insensitive to noise and is therefore very robust in scenarios such as varying ambient lighting and cross-spectral recognition. This can be attributed to the fact that IMD is a relative measure. It does not involve the values of pixel intensities directly but the difference between pixel intensities. When the lighting varies or noise is added, even though the pixel values are changed, the interval measurement is very likely to stay the same. Further illustration of the IMD robustness can be found in Section 5.

The results of encoding with IMD in application to a SWIR 1.5 m face image is provided in Figure 4. IMD operator was applied to both the original SWIR image and to the image after Gabor filtering.

### 4.4. Gabor Multiple Level Measurement

To boost the recognition performance, the complementary information of a face image encoded by four different operators is further fused up, each presenting its own level of measurement. To be specific, IMD is fused with operators at the nominal, ordinal, and ratio levels.

Figure 5 summarizes the structure of the proposed fusion approach (i.e., GMLM). An input image is first passed through a bank of Gabor filters which perform the following transformation:(9)$$\begin{array}{ccc} G(z,\theta ,s) & = & \frac{{∥K\left(\theta ,s\right)∥}^{2}}{\sigma^{2}}\;exp\;\left[-\frac{{∥K\left(\theta ,s\right)∥}^{2}{∥z∥}^{2}}{2\sigma^{2}}\right]\\ & \times & \left[e^{iK^{T}\left(\theta ,s\right)\cdot z}-e^{-\frac{\sigma^{2}}{2}}\right], \end{array}$$
where K(θ,s) is a wave vector, $\sigma^{2}$ is the variance of the Gaussian kernel, and z=(x,y) is a pixel in the input image. The magnitude and phase of the wave vector determine the scale and orientation of the oscillatory term. The wave vector can be expressed as
(10)$$K\left(\theta ,s\right)=K_{s}{\left(\cos\left(\phi_{\theta }\right),\sin\left(\phi_{\theta }\right)\right)}^{T},$$
where $K_{s}$ is the scale parameter, and $\phi_{\theta }$ is the orientation parameter. The parameters for the wave vector in the experiments of this paper are set to be $K_{s}={\left(\pi /2\right)}^{s/2}$ with s∈N and $\phi_{\theta }=\theta \pi /8$ with θ=1,2,…,8. The Gaussian kernel has a standard deviation σ=π.

The output of the Gabor filter bank is passed through a set of operators presenting four different levels of measurement: the proposed NMD at the nominal level, LBP at the ordinal level, the proposed IMD at the interval level, and WLD at the ratio level. The WLD operator [55] is defined as
(11)$$WLD_{l,N}\left(x\right)=Q_{l}\left\{\tan^{-1}\left[\sum_{i=1}^{N}\left(\frac{x_{i}-x}{x}\right)\right]\right\},$$
where $x_{i}$ is one of the *N* neighbors of a value *x* at a radius *r* (r=1,2 in the experiments). $Q_{l}$ is a uniform quantizer with l quantization levels. It should be noted that the original form of WLD is built as a combination of two operators: a differential excitation operator and a gradient orientation descriptor. In this work, only the differential excitation is engaged to encode the magnitude of the filter response. This considerably simplifies the implementation of WLD without degrading the performance of the feature extraction block.

After encoding with the operators at four levels of measurement, the encoded outputs are concatenated. The final fused operator is named Gabor Multiple Level Measurement (GMLM).

### 4.5. GMLM-CNN: The Hybrid Method

Following the encoding steps, GMLM features are applied as an input to a PCA-based succinct network, PCANet [56], which performs the final recognition task. PCANet is utilized due to its light-weight network structure, which learns informative features both easily and efficiently. Such a succinct network is quite suitable for the task of interest, that is, SWIR-Vis face verification in the presence of limited training data. The hybrid method is termed GMLM-CNN, and the overall framework of the method is shown in (see Figure 6).

The details of PCANet can be found in [56]. Here, only a general description is provided. Assuming that the number of filters in the PCA layer is L, PCA minimizes the reconstruction error within a family of orthonormal filters, i.e.,
(12)$$\underset{V}{\mathrm{argmin}}\;{∥X-VV^{T}X∥}^{2},s.t.VV^{T}=I_{L},\;$$
where X is the input training images stacked together after the removal of the mean (in our paper, it is the GMLM feature maps). $I_{L}$ is the identity matrix of size L×L. The solution $V^{T}$ is known as the *L* principal eigenvectors of $XX^{T}.$ The PCA filters are therefore expressed as
(13)$$W_{l}=mat\left(q_{l}\left(XX^{T}\right)\right),l=1,2,\ldots ,L,$$
where mat(·) is a function that maps a vector to a matrix *W*, and $q_{l}\left(XX^{T}\right)$ denotes the *l*-th principal eigenvector. The PCA layer is connected to two subsequent fully connected layers which conduct the recognition and then output the final result.

It can be clearly observed that, due to the inputs of GMLM, the proposed framework does not need elaborated convolutional layers to learn informative features as other common deep networks do. It is also worth noting that the only convolutional layer (i.e., the PCA layer) present in the network is significantly simplified due to the PCA theory-based structure.

## 5. Experiments and Analysis

### 5.1. Experimental Setup

The multispectral face dataset of Tactical Imager for Night/Day Extended-Range Surveillance (TINDERS) and Pre-TINDERS are used throughout our experiments. Both of them are collected by the Advanced Technologies Group, West Virginia High Tech Consortium (WVHTC) Foundation [57] (Figure 7).

In the experiments, we first study the performance of GMLM-CNN, the hybrid method proposed in this work. Its performance is compared to that of two other well-performing traditional methods, which are Gabor + LBP + GLBP + WLD [45] and Gabor Ordinal Measures (GOM) [58]. To acknowledge the recent progress in deep learning in application to face recognition, the performance of the above three methods is further compared to that of two typical convolutional neural network models that represent state of the art, namely Deep Coupled Learning [16] and Attribute Guided Synthesis [20].

As a side experiment, it is also investigated how each individual operator contributes to the composite operator of GMLM. We analyze their performance individually. As mentioned earlier, LBP is selected as an operator applied at the ordinal level of measurement. WLD is an operator acting at the ratio level of measurement. NMD and IMD, as introduced in this work, are the operators at the nominal level and the interval level of measurement, respectively.

Subsequently, we look into the impact of different fusion schemes on the performance. We examine different combinations of operators involving multiple levels of measurement. To logically and systematically deal with this problem, we use an incremental way of fusion. Additionally, we fuse operators at the same level to examine the scheme of same-level fusion.

Prior to encoding and matching of face images, an alignment is performed, where geometric transformations such as rotation, scaling, and translation are applied to project the eyes to a fixed position. The aligned face images are further cropped to the size of 120×112. After being cropped, images undergo an intensity normalization, where color images are converted to grayscale images and SWIR images are preprocessed using a log transformation. As mentioned earlier in this paper, during feature extraction, face images are first passed through a set of Gabor filters, followed by encoding with singular or compound operators working at different levels of measurement. To acknowledge the two-step encoding process, the results of encoding are denoted as Gabor + NMD, Gabor + IMD, Gabor + LBP, Gabor + WLD, etc.

### 5.2. Matching SWIR vs. Vis Faces by GMLM-CNN

The first experiment is conducted to match SWIR face images (the probes) to visible face images (the gallery), i.e., SWIR-VIS face verification. The verification performance of two composite operators, Gabor + LBP + GLBP + WLD and GOM, as well as two deep learning methods Deep Coupled Learning and Attribute Guided Synthesis, is compared to that of our proposed method GMLM-CNN (see Figure 8).

In the experiment, GMLM-CNN involves a set of 16 Gabor filters with two different scales and eight different orientations. The scales are set to be s=1 or 2 while the orientation angles are $\phi_{\theta }=\frac{\pi }{8}\theta ,\;\theta =1,2,\ldots ,8.$ A uniform division scheme with eight intervals is used for the IMD descriptor. Histograms with 135 bins are calculated on the encoded responses of all the four levels of measurement using non-overlapping blocks of 8×8.

The filter numbers of the PCA layer in PCANet are set to be 8, and the filter size is 5×5. The first and the second fully connected layers are of size 4096 and 96, respectively. In comparison, the Deep Coupled Learning model is built in a coupled structure with two streams, each of which corresponds to one of the heterogeneous light bands (visible light or SWIR). The Inception-ResNet v1 model [59] was chosen as the backbone. The two streams of the coupled network share initial weights and are subsequently trained to be different from each other. Attribute Guided Synthesis consists of an attribute predictor network and a U-Net generator. The two deep models are first trained on the public face dataset CASIA-WebFace [60], which includes 10,000 subjects with a total of 500,000 images. All face images are detected and aligned by an existing tool called MTCNN [61] and then cropped to the size of 120×112. The deep models are subsequently trained in a *transfer learning* way on the multispectral training subsets of Pre-TINDERS and TINDERS.

Since the training set is small in size, it is first augmented by adding synthesized noise to SWIR and Vis face images, as well as by mirroring, translating, and rotating. These actions resulted in a ten-times larger dataset (a total of 6240 images). Training of the two networks was performed on a computer equipped with an 8 Core Intel i9-9900K 3.6 GHz CPU and an Nvidia GeForce RTX 2080 Ti GPU.

As seen from Figure 8, the performance of the proposed method, GMLM-CNN, is noticeably higher than the performance of the other two composite operators, Gabor + LBP + GLBP + WLD and GOM, as well as of Deep Coupled Learning and Attribute Guided Synthesis. To be more specific, GMLM-CNN achieves GAR = 99.58% at FAR = 10% (see Table 2) followed by Gabor + LBP + GLBP + WLD and GOM with a GAR of 98.33% and 98.12%, and then by Attribute Guided Synthesis with a GAR of 97.92%. Deep Coupled Learning has the lowest GAR of 97.56%. As the FAR set to 0.1%, GMLM-CNN still achieves the highest of all (GAR = 74.37%) while GOM comes in last (GAR = 59.58%). In terms of the EER value, GMLM-CNN achieves 2.67%, which is significantly lower than the value achieved by any of the other four algorithms. Furthermore, the metrics of area under curve (AUC) and d-prime value for GMLM-CNN are also higher than that of all other methods. This demonstrates the advantage of GMLM-CNN over the other two traditional operators as well as the two deep learning methods in the case of a relatively small dataset available for training. It could be argued that the performance of Deep Coupled Learning and Attribute Guided Synthesis would be improved if a large training set was available. However, publicly available SWIR datasets such as PreTINDER and TINDER are of small sizes, and there are no other publicly available datasets of that type.

### 5.3. Performance of the Individual Measurement Level

In the previous subsection, experiments are carried out to demonstrate the advantage of using GMLM as a composite operator. It is not evident, however, how the inclusion of each individual level of measurement affects performance of the cross-spectral face recognition system. It would be interesting to rank the levels in terms of their contribution to the performance. Therefore, this subsection presents the results of a different experiment, where each of the four individual operators contributing to GMLM is used individually when encoding the input face. The performance of the four individual operators is summarized in Table 3.

From Table 3, it can be concluded that Gabor + IMD yields the best performance of EER = 4.33%, while Gabor + NMD displays the worst performance in terms of EER. Its EER is equal to 8.39%. The second most important component is the ordinal level, followed by the ratio level. The EER values of Gabor + LBP and Gabor + WLD are 5.99% and 7.92%, respectively. The AUC and d-prime value for Gabor + IMD are also the highest. This suggests that the interval level of measurement is the most informative in terms of cross-spectral face verification performance. The second important component is the ordinal level, and the third in order of importance is the ratio level. The nominal level contributes the least.

This observation contradicts a natural yet naive expectation that the most complex operator should display the best performance. It can be explained in the sense that the ordinal and interval levels encode information of a "looser" relationship between neighboring pixels rather than an "exact" relationship as the ratio level does, which in turn makes the two levels more robust with respect to the cross-spectral verification scenarios where heterogeneous pixel values vary significantly (in other words, the exact relationship is unstable). However, analyzing the performance in Table 3 and observing how closely they follow one another leads to a natural conclusion that every level is important and each level contributes to the performance of GMLM.

### 5.4. Performance of Fused Measurement Levels

In this subsection, the impact of different fusion combinations on the recognition performance is further investigated.

#### 5.4.1. Fusing Complementary Levels

One would naturally argue that, since each of the four levels of measurement contributes to the performance, all of them are necessary. This argument is indeed true and is supported by our experimental results shown below. For instance, when GOM is replaced with Gabor + LBP + WLD + GLBP, the recognition performance is boosted due to inclusion of the ratio level of measurement in addition to the ordinal level. To abbreviate the notation, we will call the combination scheme of Gabor + LBP + WLD + GLBP as 2nd + 4th levels (or 2 + 4 in short). In total, four different fusion combinations in an *incremental* manner are considered, as summarized in Table 4.

As can be seen from Table 5, when the algorithm is changed from GOM to Gabor + LBP + WLD + GLBP, the GAR value is increased from 98.12% to 98.33% at FAR = 10% and EER is decreased from 4.94% to 3.86%.

When the combination is further changed to Gabor + LBP + NMD + WLD (i.e., the 1st level of measurement is further added), the GAR value increases to 98.58% and EER decreases to 3.66%. Finally, when IMD, i.e., the 3rd level, is added to Gabor + LBP + NMD + WLD, the performance reaches the highest value of GAR = 99.58% and achieves the lowest value of EER = 2.6%, yielding the best result of the experiment.

To summarize, when an operator at a measurement level is incrementally combined with other measurement levels, the performance gradually improves. This conclusion supports the statement made earlier in this paper that different levels of measurement contain complementary information useful for the face recognition task.

#### 5.4.2. Fusing Same-Level Operators

However, similar to the problem of selecting informative features [62,63,64], simply involving a larger number of operators does not necessarily lead to improved performance, especially when several operators in the combination represent the same level of measurement. This is due to correlation and redundancy of information extracted by the operators acting at the same level of measurement. Information redundancy rarely improves verification performance.

In order to demonstrate this, a new experiment is designed where two operators both representing the 4th level are combined. The performance of such a combination is compared with that of a single 4th-level operator. The 4th level is chosen for demonstration because of a large number of operators in the literature acting at this level. In particular, WLD and HOG are selected. The results listed in Table 6 demonstrate that GAR and EER values do not necessarily improve when fusing the two operators acting at the same level of measurement. As a matter of fact, both GAR and EER become even worse when the same level of measurement (note that both HOG and WLD are at the fourth level) is added. This can be attributed to the fact that operators performing at the same level extract similar information from the data and therefore introduce redundancy.

From the results of different operator-fusing experiments in this section, it is concluded that the fusion of operators representing different levels of measurement is more beneficial than fusion of operators representing the same level. Therefore, our design of GMLM which follows such a fusing rule can be justified and explained.

## 6. Conclusions

This paper focuses on the problem of cross-spectral face verification where face images of a relatively new IR subband, SWIR, are matched against images acquired in the visible light spectrum. The proposed hybrid solution takes advantage of both traditional feature engineering and modern deep learning techniques to overcome the issue of limited training data. Both fusion among handcrafted features and fusion between handcrafted and deep-learned features are involved.

Two new individual operators, the Nominal Measurement Descriptor (NMD) and the Interval Measurement Descriptor (IMD), are firstly introduced, representing the nominal and interval levels of measurement, respectively. Then, operators with respect to all the four levels of measurement are fused, resulting in a new composite operator named Gabor Multiple Level Measurement (GMLM). Such fusion extracts complementary information and yields considerably improved cross-spectral face recognition performance. Finally, the fused features of GMLM are passed through a simple-structured but efficient PCA-based neural network which selects informative features and also performs recognition. The overall framework is named GMLM-CNN. Throughout the experiments, transfer learning and data augmentations are utilized. Experimental results show that GMLM-CNN outperforms two other well-performing composite operators as well as the state-of-the-art deep learning-based methods, Deep Coupled Learning and Attribute Guided Synthesis.

Contribution of each individual operator representing four distinct levels of measurement is also investigated. It is observed that the proposed interval level (IMD) contributes the most to the performance. Finally, effects of different fusion schemes are also studied in detail. It is concluded that fusing operators at complementary levels of measurement is beneficial to the performance, while fusing at the same level results in little performance improvement or even leads to degraded performance.

For future work, we plan to replace PCANet with matrix decomposition-based networks that are even more efficient and succinct, such as SVD decomposition based networks and LU decomposition based networks. In addition, rather than a hybrid solution, a pure deep learning solution can be proposed where the hand-crafted composite operator of GMLM can be also replaced by pure neural networks inspired by the measurement theory. For instance, special convolutions can be designed to comply with the operations respective to different measurement levels.

## Figures and Tables

**Figure 1 sensors-22-09500-f001:**
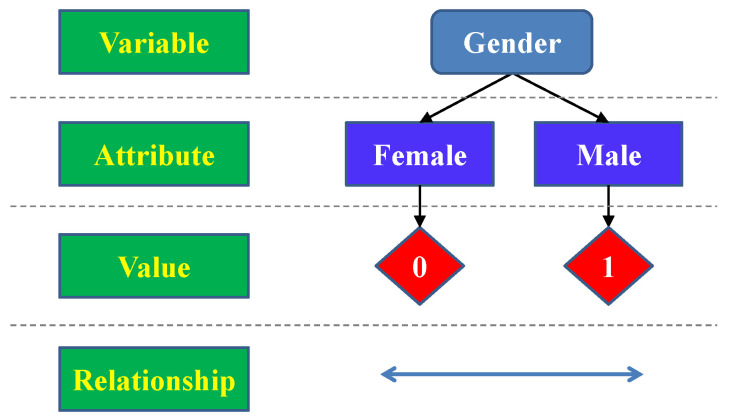
Example of nominal measurement in gender classification.

**Figure 2 sensors-22-09500-f002:**
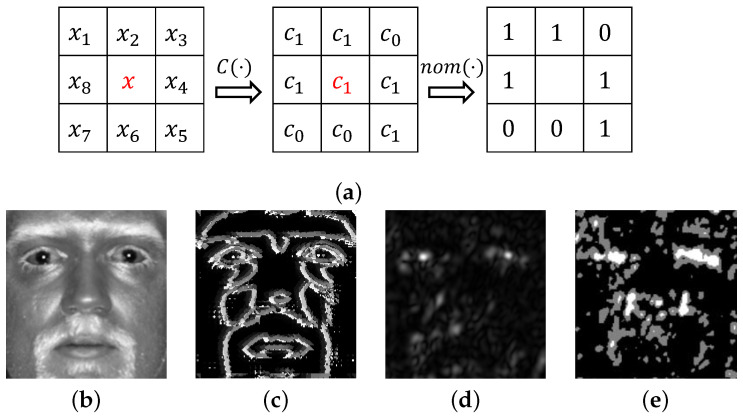
Encoding with NMD. (**a**) From left to right: the input, the output after class assignment and the final nominal measurement values; (**b**) a cropped SWIR face image with contrast adjustment; (**c**) NMD encoding applied to the face image in (**b**); (**d**) the magnitude response of the Gabor filter applied to the cropped face; (**e**) NMD encoding applied to Gabor features in (**d**).

**Figure 3 sensors-22-09500-f003:**
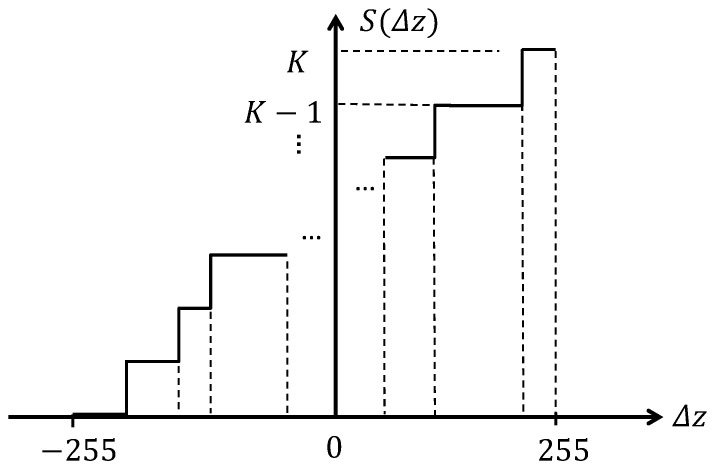
Illustration of IMD with non-uniform partition schemes.

**Figure 4 sensors-22-09500-f004:**
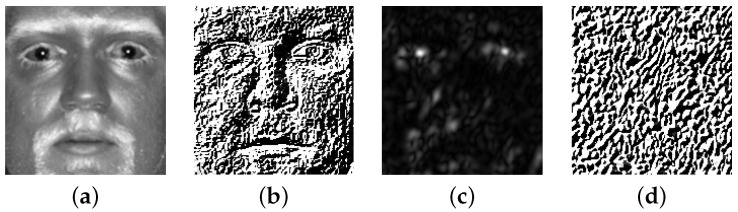
Encoding with IMD: (**a**) a cropped SWIR face image; (**b**) the result of applying IMD to the cropped face in (**a**); (**c**) the magnitude response of the Gabor filter; (**d**) IMD encoding applied to the Gabor filtered image in (**c**).

**Figure 5 sensors-22-09500-f005:**
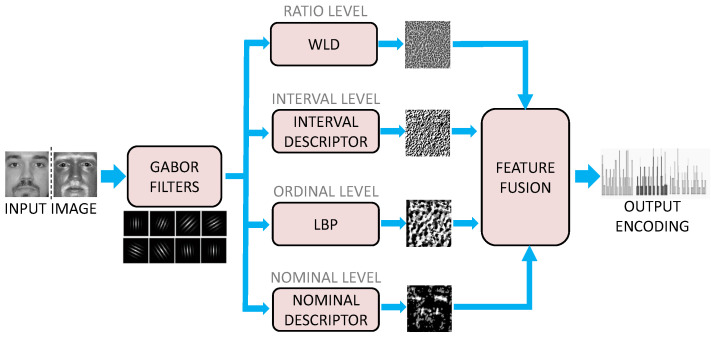
The structure of the fused operator proposed in the paper: Gabor Multiple Level Measurement.

**Figure 6 sensors-22-09500-f006:**
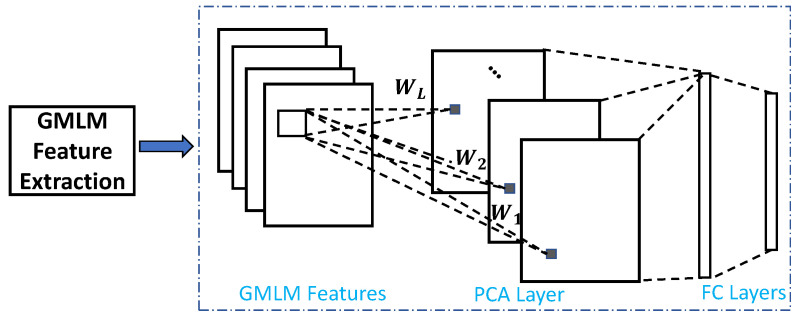
The overall structure of GMLM-CNN, the proposed hybrid method: GMLM feature extraction first and then PCANet Learning.

**Figure 7 sensors-22-09500-f007:**
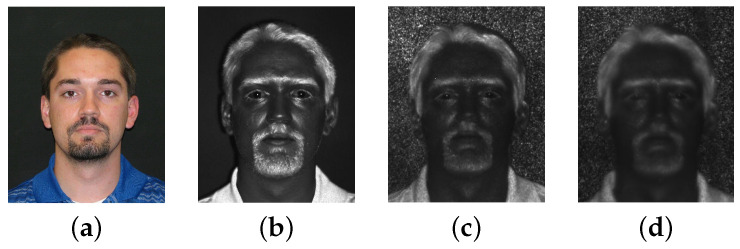
Sample face images: (**a**) visible light; (**b**–**d**) SWIR at 1.5 m, 50 m, and 106 m, respectively.

**Figure 8 sensors-22-09500-f008:**
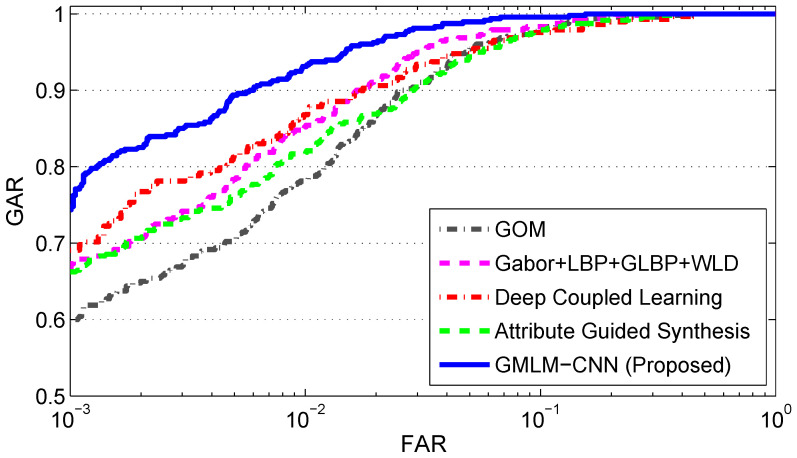
ROC of Matching SWIR against visible light images.

**Table 1 sensors-22-09500-t001:** Summary of the four levels of measurement with example operators for face recognition. Note that NMD and IMD (marked with asterisks) are new operators proposed in this paper.

Measurement	Arithmetic and Logical Operations	Complexity	Meaning	Example Operators
nominal	=, ≠	lowest	categories	NMD *
ordinal	=, ≠, <, >	medium	orders	LBP
interval	=, ≠, <, >, +, −	medium	distance meaningful	IMD *
ratio	=, ≠, <, >, +, −, ×, ÷	highest	absolute zero meaningful	WLD, HOG

**Table 2 sensors-22-09500-t002:** Comparison of GARs and EER between different methods: SWIR vs. visible light.

Method	GAR (%) at FAR = $10^{-1}$	GAR (%) at FAR = $10^{-3}$	EER (%)	AUC	d-Prime
GOM [58]	98.12	59.58	4.94	0.9896	2.92
Gabor + LBP + GLBP + WLD [45]	98.33	66.67	3.86	0.9929	2.98
Deep Coupled Learning [16]	97.56	69.09	4.93	0.9898	2.86
Attribute Guided Synthesis [20]	97.92	66.25	5.36	0.9895	2.80
**GMLM-CNN** **(Proposed)**	**99.58**	**74.37**	**2.67**	**0.9971**	**3.13**

**Table 3 sensors-22-09500-t003:** GARs and EER for each individual measurement level: SWIR vs. visible light.

Method	Measurement Level	GAR (%) at FAR = $10^{-1}$	GAR (%) at FAR = $10^{-3}$	EER (%)	AUC	d-Prime
Gabor + NMD	Nominal	92.92	42.92	8.39	0.9713	2.49
Gabor + LBP	Ordinal	95.42	52.92	5.99	0.9776	2.73
**Gabor + IMD**	**Interval**	**97.92**	**65.83**	**4.33**	**0.9918**	**2.91**
Gabor + WLD	Ratio	93.13	53.13	7.92	0.9763	2.69

**Table 4 sensors-22-09500-t004:** Summary of different combination schemes.

Fusion Scheme	Levels	Notation
GOM	Ordinal	2
Gabor + LBP + GLBP + WLD	Ordinal + Ratio	2 + 4
Gabor + LBP + NMD + WLD	Nominal + Ordinal + Ratio	1 + 2 + 4
GMLM-CNN	Nominal + Ordinal + Interval + Ratio	1 + 2 + 3 + 4

**Table 5 sensors-22-09500-t005:** GARs and EER when fusing complementary levels.

Fusion Scheme	Notation	GAR (%) at FAR = $10^{-1}$	GAR (%) at FAR = $10^{-3}$	EER (%)	AUC	d-Prime
GOM	2	98.12	59.58	4.94	0.9896	2.92
Gabor + LBP + GLBP + WLD	2 + 4	98.33	66.67	3.86	0.9929	2.98
Gabor + LBP + NMD + WLD	1 + 2 + 4	98.58	69.37	3.66	0.9953	2.95
**GMLM-CNN**	**1 + 2 + 3 + 4**	**99.58**	**74.37**	**2.67**	**0.9971**	**3.13**

**Table 6 sensors-22-09500-t006:** GARs and EER when fusing operators at the same level.

Method	GAR (%) at FAR = $10^{-1}$	GAR (%) at FAR = $10^{-3}$	EER (%)	AUC	d-Prime
Gabor + WLD	93.13	53.13	7.92	0.9763	2.69
Gabor + WLD + HOG	93.12	51.67	8.13	0.9761	2.62

## Data Availability

No new data were created or analyzed in this study. Data sharing is not applicable to this article.

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
