# Peer review of "GMLM-CNN: A Hybrid Solution to SWIR-VIS Face Verification with Limited Imagery"

_sensors, 2022, doi:10.3390/s22239500_

Round 1

Reviewer 1 Report

Minor change suggestions for the paper

Line

Comment

12,47,207,225,234, ….

Too many use of  “we” for the authors mentioned  in the paper, passive centences will be better

If its OK for editors neglect the comment

47

“SWIR for study due to its advantages  over other IR subbands such as NIR and LWIR”
Any references for this comment, references are not good for that comment or missing

Reference 24 is not valid

http://www.sensorsinc.com/facilitysecurity.html

no page available

Reference 30 is not valid

http://www.cognitec-systems.de

no page available

Reference 25 is not valid

“Goodrich,Lightweight SWIR Sensor for Target Detection on Board UAV Equipment”

What is this reference, book, web page ?

307-312

Example is good but not necessary for research paper,

341

“Consider the 8-pixel neighborhood of a pixel”

Change the wording, this is not a lecture book.

445

“In our experiments..”

Use passive centences,

Author Response

We greatly thank the first Reviewer for his/her valuable time and insightful suggestions provided in the review summary.  We accept all the suggestions and have revised the draft point by point, which we attach for your consideration. Modifications made in the revised manuscript are highlighted in color (grammar mistakes, typos and misspellings are marked in red while all other changes are in blue).

The following modifications and improvements have been applied to the paper as suggested.

  1. The sections of Introduction and Review of Relevant Research are improved.
  2. All equations are double-checked to be technically correct.
  3. The language of the paper is refined by a native English speaker who holds a PhD degree in engineering. Grammar mistakes, typos and misspellings are all corrected.
  4. Figures are improved.
  5. Additional performance metrics are included as well as the future work of this paper.
  6. Potential applications of the proposed method are also discussed.
  7. The frequent usage of “we” is now avoided by changing the sentences to passive voice.
  8. Several references that have missing information are now updated.
  9. Redundant examples and inappropriate tone of words are either removed or rephrased.

A point-to-point answer to the comments:

Point 1: Line 12,47,207,225,234, …. Too many uses of “we” for the authors mentioned in the paper, passive sentences will be better.

Response 1:  We thank the reviewer for this suggestion. The frequent usage of “we” is now avoided by changing the sentences to passive voice.

Point 2: “SWIR for study due to its advantages over other IR subbands such as NIR and LWIR” Any references for this comment, references are not good for that comment or missing

Response 2: Thanks for pointing out this problem. New references are now added, and old references that are problematic have been corrected.

Point 3: Reference 24 is not valid: http://www.sensorsinc.com/facilitysecurity.html  no page available.

Reference 30 is not valid: http://www.cognitec-systems.de   no page available

Reference 25 is not valid: “Goodrich,Lightweight SWIR Sensor for Target Detection on Board UAV Equipment” What is this reference, book, web page ?

Response 3: Thank for the comment. We have now updated all the three problematic references by adding the missing information.

Point 4: Line 307-312 Example is good but not necessary for research paper,

Response 4: Indeed, such an elaborate example is unnecessary. We take the suggestion and avoid it. Words are rephrased.

Point 5: Line 341,Consider the 8-pixel neighborhood of a pixel” Change the wording, this is not a lecture book.

Response 5: We take the suggestion and change to tone of this sentence.

Point 6: Line 445,In our experiments..” Use passive sentences,

Response 5: Thanks again for pointing out this issue. Now the sentence has been changed to passive voice.

Sincerely,

Liaojun Pang

Reviewer 2 Report

Authors drafted well about the combination of feature engineering and modern deep learning techniques for face verification with limited imagery. This is very useful in limited imagery as encountered in the SWIR band.  The author not only demonstrates the advantages of this method in theory, but also verifies the effectiveness of the improved method through experiments.
I have some comments as follows:

1. Introduction part can be improved and elaborated with objectives of the proposed system

2. The literature review made good and adequate, but recent reference articles in imagery verification from the reputed journals must be included.

3. Drawbacks of the existing systems from the literature review made have to be discussed.

4. I came across many Equations in this article, kindly ensure the equations
mentioned are described in proper format.

5. I recommend authors to minimize the grammatical errors which I came across while reading the manuscript.

6. Image given as figure 5 must be clear and I suggest authors to enhance the functional components in it.

7. Other performance measures can also be included. Also include the Future work of the proposed model.

8. Applications of the proposed model to be mentioned.

Author Response

Response to the Comments by Reviewer 2

We greatly thank the first Reviewer for his/her valuable time and insightful suggestions provided in the review summary.  We accept all the suggestions and have revised the draft point by point, which we attach for your consideration. Modifications made in the revised manuscript are highlighted in color (grammar mistakes, typos and misspellings are marked in red while all other changes are in blue).

The following modifications and improvements have been applied to the paper as suggested.

  1. The sections of Introduction and Review of Relevant Research are improved.
  2. All equations are double-checked to be technically correct.
  3. The language of the paper is refined by a native English speaker who holds a PhD degree in engineering. Grammar mistakes, typos and misspellings are all corrected.
  4. Figures are improved.
  5. Additional performance metrics are included as well as the future work of this paper.
  6. Potential applications of the proposed method are also discussed.
  7. The frequent usage of “we” is now avoided by changing the sentences to passive voice.
  8. Several references that have missing information are now updated.
  9. Redundant examples and inappropriate tone of words are either removed or rephrased.

A point-to-point answer to the comments:

Point 1: Introduction part can be improved and elaborated with objectives of the proposed system

Response 1:  We thank the reviewer for this suggestion. The new section of Introduction has been extended with objectives. See the two paragraphs from line 75 to 91.

Point 2: The literature review made good and adequate, but recent reference articles in imagery verification from the reputed journals must be included.

Response 2: Thanks for pointing out this problem. New references are now added.

Point 3: Drawbacks of the existing systems from the literature review made have to be discussed.

Response 3: Thank for the suggestion. We summarize the drawbacks of both traditional methods (from line 178 to 185) and deep learning methods (see line 225 to 239).

Point 4: I came across many Equations in this article, kindly ensure the equations mentioned are described in proper format.

Response 4: Thanks for reminding of the technical correctness requirement of the equations. We double checked all the equations to be correct in the revised manuscript.

Point 5: I recommend authors to minimize the grammatical errors which I came across while reading the manuscript.

Response 5: Sorry for the language issue in the 1st submission. The language of the paper is now refined by a native English speaker who holds a PhD degree in engineering. Grammar mistakes, typos and misspellings are all corrected.

Point 6: Image given as figure 5 must be clear and I suggest authors to enhance the functional components in it.

Response 6: We take the suggestion and redraw Figure 5 which now provides more information about the functional components.

Point 7: Other performance measures can also be included. Also include the Future work of the proposed model.

Response 7: We take the suggestion and have added two more performance metrics: area under curve (AUC) and d-prime value. See the tables for updates.

Point 8: Applications of the proposed model to be mentioned.

Response 8: This is a good suggestion, especially for the interest of the readers. See the discussion about potential applications from line 48-53.

Sincerely,

Liaojun Pang
